# Whole Exome Sequencing as a Diagnostic Tool for Unidentified Muscular Dystrophy in a Vietnamese Family

**DOI:** 10.3390/diagnostics10100741

**Published:** 2020-09-24

**Authors:** Ngoc-Lan Nguyen, Can Thi Bich Ngoc, Chi Dung Vu, Thi Thu Huong Nguyen, Huy Hoang Nguyen

**Affiliations:** 1Graduate University of Science and Technology, Vietnam Academy of Science and Technology, 18 Hoang Quoc Viet str., Cau Giay, Hanoi 100000, Vietnam; lannguyen@igr.ac.vn (N.-L.N.); nguyenthithuhuong@hdu.edu.vn (T.T.H.N.); 2Institute of Genome Research, Vietnam Academy of Science and Technology, 18 Hoang Quoc Viet str., Cau Giay, Hanoi 100000, Vietnam; 3Center for Rare Diseases and Newborn Screening, Department of Endocrinology, Metabolism and Genetics, Vietnam National Children’s Hospital, 18/879 La Thanh str., Dong Da, Hanoi 100000, Vietnam; ngocctb@nch.org.vn (C.T.B.N.); dungvu@nch.org.vn (C.D.V.)

**Keywords:** LAMA2-related muscular dystrophy, WES, *LAMA2* variants, Vietnamese, c.778C>T, c.2987G>A

## Abstract

Muscular dystrophies are a group of heterogeneous clinical and genetic disorders. Two siblings presented with characteristics like muscular dystrophy, abnormal white matter, and elevated serum creatine kinase level. The high throughput of whole exome sequencing (WES) makes it an efficient tool for obtaining a precise diagnosis without the need for immunohistochemistry. WES was performed in the two siblings and their parents, followed by prioritization of variants and validation by Sanger sequencing. Very rare variants with moderate to high predicted impact in genes associated with neuromuscular disorders were selected. We identified two pathogenic missense variants, c.778C>T (p.H260Y) and c.2987G>A (p.C996Y), in the *LAMA2* gene (NM_000426.3), in the homozygous state in two siblings, and in the heterozygous state in their unaffected parents, which were confirmed by Sanger sequencing. Variant c.2987G>A has not been reported previously. These variants may lead to a change in the structure and function of laminin-α2, a member of the family of laminin-211, which is an extracellular matrix protein that functions to stabilize the basement membrane of muscle fibers during contractions. Overall, WES enabled an accurate diagnosis of both patients with *LAMA2*-related muscular dystrophy and expanded the spectrum of missense variants in *LAMA2*.

## 1. Introduction

The muscular dystrophies (MDs) are a heterogeneous group of predominantly inherited conditions characterized by progressive muscle weakness with fiber degeneration [1,2,3]. The estimated prevalence of MDs in Assiut Governorate, Egypt was 26.8 per 100,000 [4]. Based on clinical findings, MDs are divided into six groups—Duchenne and Becker MDs, distal myopathies, limb-girdle MD, facio–scapulo–humeral muscular dystrophy, myotonic dystrophy, and congenital muscular dystrophies (CMDs) [5]. A novel way to classify MDs is established due to protein classes such as the extracellular matrix and basement membrane proteins (collagen 6, laminin 211, and the cellular receptor α-dystroglycan), the sarcolemma-associated proteins (complex of dystrophin, sarcoglycans, and dystroglycan, sarcoglycanopathies, dysferlin, and anoctamin 5), the enzymes or proteins with putative enzymatic function (the glycosylation of α-dystroglycan and calpain 3), the nuclear membrane proteins (lamin A/C, emerin, nesprin 1 and 2, LUMA, matrin 3), the sarcomeric proteins (titin and myosin), the endoplasmic reticulum proteins (selenoprotein N1 and TRAPPC11), and the other proteins [2].

CMDs represent a small group of MDs and are characterized by hypotonia and weakness with the onset at birth or in infancy [5,6,7]. The different types of CMDs can be distinguished in several different ways [5]. In summary, CMDs consist of CMD with merosin deficiency, Fukuyama CMD, Ullrich CMD, Bethlem myopathy, rigid spine syndrome, Walker–Warburg syndrome, CMD due to ITGA7 deficiency, CMD with hypoglycosylation of dystroglycan or fatty liver and infantile-onset cataract caused by TRAPPC11 mutations or joint hyperlaxity or mitochondrial structural abnormalities or rigid spine related to ACTA1 or cataracts and intellectual disability, CMD related to LMNA or dynamin 2 or telethonin or MSTO1 or RYR1, GOLGA2-related congenital muscle dystrophy with brain involvement, MD congenital Danignon–Chauveau type, muscle–eye–brain disease, and muscular dystrophy–dystroglycanopathy [8].

CMDs were estimated to have a prevalence of 0.563 per 100,000 in Italy [9] and 0.017–0.083 per 100,000 in China [10]. Laminin α-2 related congenital muscular dystrophy was the most common group of CMDs in Europe [11,12], UK [13], Denmark [14], China [10], and Qatar [15]. The human laminin α-2 gene (*LAMA2*; #156225) is located at the chromosomal locus 6q22.33, consists of 65 exons, and encodes the 3,122 amino acids of laminin α-2 subunit [12]. The laminin α-2 subunit is a part of the laminin-211 (α-2, β-1, and γ-1 subunits) and the laminin-221 (α-2, β-2, and γ-1 subunits), which attach to each other or other proteins like other laminins, nidogen, and agrin in the extracellular matrix and in the membrane of muscle cells to maintain the stability of muscular fibers [16]. The laminin-211 and laminin-221 play a role in the skeletal muscles. The two major receptors of laminin-211 and laminin-221 in the skeletal muscle are α-dystroglycan and α7β1 integrin [17]. *LAMA2* mutations cause deficiency or absence of the laminin α-2 subunit, leading a lack of laminin-211 and laminin-221, resulting a reduced strength and stability of skeletal muscle tissue. Mutations in *LAMA2* have been reported to be associated with *LAMA2*-related muscular dystrophy or merosin deficiency as LAMA2 is also called merosin. This disorder follows an autosomal recessive mode of inheritance. Diagnosis of *LAMA2*-related muscular dystrophy is based on clinical findings, elevated serum creatine kinase, abnormal white matter signal on T2-weighted MRI, complete or partial laminin α2 subunit on IHC staining of muscle, and biallelic mutation of the *LAMA2* gene [18]. Molecular diagnosis of *LAMA2*-related muscular dystrophy provides the advantage of avoiding invasive muscle biopsy [19,20] and allows determination of the inheritance pattern. DNA testing is performed for other family members when the mutation in the proband is known. Recently, Saredi et al. [21] suggested WES might be an efficient method to reach a diagnosis of *LAMA2*-related muscular dystrophy.

Approximately 587 genes have been suggested as disease genes for neuromuscular disorders, and out of those, 69 genes are responsible for MDs and CMDs [8]. MDs and CMDs are caused by changes in certain genes encoding the proteins providing strength to the muscle structure. Clinical heterogeneity was observed between and within the types and subtypes of MDs [2,22,23] and CMDs [2,5,6,7]. Fukuyama CMD showed clinical overlap with muscle–eye–brain disease like inability to walk, mental retardation, ocular involvement, and lissencephaly type II [24]. O’Grady et al. [7] selected 113 CMD patients based on clinical presentations like muscle weakness and hypotonia within the first 2 years of life, delayed gross motor milestones, contractures or scoliosis, dystrophic changes. The authors also excluded patients suspected to have congenital myopathy. However, molecular analyses confirmed only 76% of patients to be associated with CMDs, while 19% of patients were related with congenital myopathies. These findings reflect the clinical overlapping between CMDs and congenital myopathies [7]. Late-onset *LAMA2*-related MD patients may also present clinical features similar to those of a childhood-set limb-girdle MD such as proximal muscle weakness and delayed motor milestones [25]. Mutations in the same gene can cause different types of MDs. For example, mutations in the *LMNA* gene can result both limb-girdle muscular dystrophy [26] and Emery–Dreifuss muscular dystrophy. Mutations in the *POMT2* gene can cause severe brain involvement like Walker–Warburg syndrome [27] and muscle–eye–brain disease [28] as well as limb-girdle muscular dystrophy [29]. Mutations in collagen 6α led to severe Ulrich CMD or milder Bethlem myopathy [24]. MDs showed genetic heterogeneity by sharing different clinical manifestations with the mutations in different genes, such as nine known genes associated with Emery–Dreifuss muscular dystrophy [30] and twenty-six known genes associated with limb-girdle muscular dystrophy [31]. Due to overlapping clinical symptoms and the multiple possible genetic causes for MDs and CMDs, it is difficult to obtain an accurate diagnosis for patients [23,24].

Next-generation sequencing is merging as a promising tool for diagnosis of MDs and CMDs. Indeed, next-generation sequencing has been applied in the investigation of the genetics of unidentified MDs and achieved valuable results [7,32,33,34]. For example, Dardas et al. [32] applied whole exome sequencing (WES) and obtained diagnosis in seven out of eight unrelated consanguineous Jordanian families with MDs. WES is also a powerful diagnostic tool for limb-girdle muscular dystrophy [35,36]. Moreover, it is evident that many additional genes associated with MDs and CMDs are being discovered every year [8,37]. Moreover, WES data can be reanalyzed in some undiagnosed cases to obtain a definitive diagnosis.

In this report, we present clinical presentations and molecular analysis of two siblings suspected with muscular dystrophies. Due to numerous causative genes with X-linked, autosomal dominant/recessive inheritances, WES was performed for two siblings and their parents to rapidly detect the disease-causing variants as well as to reduce both cost and time.

## 2. Case Presentation

### 2.1. Patients

The study was performed according to the Declaration of Helsinki (2013) and approved by the Ethics Committee of the Institute of Genome Research (No. 16/QD-NCHG on 22 March 2018, Institute of Genome Research Institutional Review Board, Hanoi, Vietnam). Here, we studied in a family of Ksingmul–an ethnic minority group–in the Son La province, Vietnam. A written informed consent was obtained from both parents for genetic analyses and publication of this case report.

The presentation of the two siblings were similar (Table 1). The family history could not be explored as the father had been adopted. Patient 1, the first child in the family, is a male born at full-term and by a vaginal delivery. The birth weight was not recorded. The mother’s obstetric history was normal. The child was completely healthy after birth, but he was never able to roll over in the first year of life. He could sit at age one year and stand at age four years. He achieved independent walking at age five years and began toe walking at age six years. He was referred to the Vietnam National Children’s Hospital at the age of six years with a chief complaint of motor retardation. Clinical manifestations included myopathic face, open mouth, prominent jaw, bifid uvula, distal contractures of the fingers, a wide-based stance, hyperlordosis, muscular atrophy, macroglossia, motor delay, and a positive Gowers’ sign. No pseudohypertrophy of the tongue or calf muscles was found. He had a normal verbal cognition and no history of seizure. Brain magnetic resonance imaging (MRI) investigations revealed diffuse white matter changes. The white matter of bilateral hemisphere and both posterior internal capsules were diffusely hyperintense on T2W and fluid-attenuated inversion recovery (FLAIR) sequences, hypointense on T1W sequence, but not restricted on diffusion with symmetric appearance. The corpus callosum was normal in shape, size, and signal. The ventricular system was normal in shape and not dilated. The midline was not shifted. No fluid collection in meningeal space was detected. Size of the pituitary gland was normal. No abnormality was found in pontocerebellar angle. Medulla, pons and high cervical spinal cord were normal in structure and signal. Cerebral nerves were normal. No abnormality of paranasal and mastoid sinuses was found. The cardiac ultrasound was normal. The blood tests showed elevated creatine kinase of 942 UI/L and normal levels of alanine aminotransferase (ALT) and aspartate aminotransferase (AST). He had normal respiratory and digestive systems. The blood karyotype was 46, XY. At the age of 9, he could not walk.

Patient 2 is a female and the second child in the family. She was born at term by a vaginal delivery. No measurement of her birth weight was available. The mother’s pregnancy was normal. The child was completely healthy after birth but could not sit until one year of age or walk until four years of age. She started toe walking at five years of age. She presented similar phenotype with her elder brother (Patient 1), except a broad uvula (Table 1).

### 2.2. Whole Exome Sequencing, Variant Calling and Annotation

Analyses of the *DMD*, *SMN* and *GAA* genes of two siblings were normal; therefore, we applied WES to detect pathogenic variants. WES was performed for two siblings and their parents. Genomics DNA was extracted from total blood samples using the QIAGEN Dneasy Blood and Tissue Kit (Qiagen, Valencia, CA, USA). All DNA samples were stored at −80 °C for exome sequencing. WES and annotation were carried as previously described [38].

WES yielded 9.26–11.36 Gb data with 86.6–93.7% of target covered over 30×, indicating good quality reads for further analyses (Appendix A). A total of 95,859–97,062 single nucleotide polymorphisms and 13,626–14,332 insertions/deletions were detected in each member of the family.

### 2.3. Filtration of Variants

Candidate variants were filtered by the following criteria: prioritization genes, frequency, the effect of variants, in-house database, and the pattern of inheritance. Variants located in the 587 candidate genes associated with neuromuscular disorders (the 2020 version of the gene table of neuromuscular disorders) were screened. The frequencies of variants were checked from 1000 Genome (http://browser.1000genomes.org). Rare variants (allele frequency < 0.001) with high and moderate impact (truncating variants, frameshift/inframe insertions and deletions, and missense variants) were selected. Missense variants which were predicted as tolerance in Sorting Intolerant from Tolerant (SIFT; http://sift.jcvi.org/) and benign in Polymorphism Phenotyping v2 (PolyPhen-2; http://genetics.bwh.harvard.edu/pph2/) were excluded. Variants occurring in the in-house database were also removed. The zygosity of the remaining variants was considered and compared with dominant, recessive, or X-linked modes of inheritance of the genes. Variants were de novo in the dominant genes and were homozygous in patients and heterozygous in their parents; in the recessive genes, they were selected as disease-causing. The candidate variants were manually inspected for their WES-read alignments using the Integrative Genomics Viewer v.2.6.3 (IGV, http://software.broadinstitute.org/software/igv/). The nomenclature of the identified candidate variants was given according to human genome variation society guidelines. Nucleotide numbering for *LAMA2* is according to the reference transcript NM_000426.3. C1 is the A of the ATG translation initiation codon and p.M1 is the initiation codon.

A total of 3405–4622 variants mapping to genes associated with neuromuscular disorders were identified in the family (Appendix A). Of those, twenty-four rare and impacted variants in 19 genes were identified in two siblings (Appendix A). These variants were inherited from their parents. No de novo possible pathogenic variants were detected in both siblings. Only two missense variants, c.778C>T in exon 5 and c.2987G>A in exon 21, of the *LAMA2* gene, were identified as possible pathogenic variants in two siblings. These two variants were homozygous in two siblings and heterozygous in their parents. Using IGV inspection, the two candidate variants were confirmed and selected due to the reliable coverage and the read depth (Figure 1).

### 2.4. Sanger Sequencing for Variant Confirmation and Familial Segregation Analysis

To confirm states of candidate variants, two primer sets were designed using the Primer BLAST (https://www.ncbi.nlm.nih.gov/tools/primer-blast). The forward primer LAMA2-5F (5′-tgcgtaactaattgggagaatgg-3′) and reverse primer LAMA2-5R (5′-ggcacagcttggatctgaaca-3′) were used to amplify exon 5 of *LAMA2*. Whereas exon 21 of *LAMA2* was amplified using the forward primer LAMA2-21F (5′-gttcctctgttcccacctga-3′) and reverse primer LAMA2-21R (5′-cgttgtatcaatctgtgcttcc-3′). Amplification and Sanger sequencing were performed as previously described [38]. Sequencing results were analyzed using BioEdit (Ibis Biosciences, Carlsbad, CA, USA) and compared with the *LAMA2* gene sequence from the NCBI database (NM_000426.3).

Sanger sequencing results confirmed the presence of two variants, c.778C>T and c.2987G>A, in the homozygous state, in two patients, as well as the carrier status of the parents (Figure 2a). Such results indicated that two siblings inherited mutant alleles from both of their parents. The variant c.778C>T substituted histidine at the residue 260 with tyrosine (p.H260Y). The variant c.2987G>A changed from cysteine to tyrosine at the residue 996.

### 2.5. In Silico Analyses and Pathogenic Interpretation

The pathogenicity of candidate variants was further evaluated by in silico analyses using Align-GVGD (http://agvgd.hci.utah.edu/agvgd_input.php), Combined Annotation Dependent Depletion (CADD; https://cadd.gs.washington.edu/snv), Functional Analysis through Hidden Markov Models (FATHMM; http://fathmm.biocompute.org.uk/inherited.html), Mutation Assessor (http://mutationassessor.org/r3/), Mutation Taster (http://www.mutationtaster.org/), PANTHER (http://www.pantherdb.org/tools/hmmScoreForm.jsp), Predictor of human Deleterious Single Nucleotide Polymorphisms (PhD-SNP, https://snps.biofold.org/phd-snp/phd-snp.html), Pathological Mutations on proteins (Pmut; http://mmb.irbbarcelona.org/PMut/analyses/new), PON-P2 (http://structure.bmc.lu.se/PON-P2), Protein Variation Effect Analyzer (PROVEAN; http://provean.jcvi.org/index.php), SNPs & GO (https://snps.biofold.org/snps-and-go/snps-and-go.html), SNAP2 (https://www.rostlab.org/services/snap/), and UMD-Predictor (http://umd-predictor.eu/) prediction programs. The candidate variants were classified according to the American College of Medical Genetics and Genomics (ACMG) guidelines [39].

Variant c.778C>T (p.H260Y) is not novel (rs780568352), but is a very rare mutation observed in the heterozygous state with allele frequency accounting for 4 out of 250,644 exomes in the Genome Aggregation Database (gnomAD) and 3 out of 120,950 exomes in the Exome Aggregation Consortium (ExAC), probably being asymptomatic carriers. This variant was predicted to be damaging in PolyPhen-2, disease causing in Mutation Taster, possibly damaging in PANTHER, pathogenic in Align-GVGD, MutPred and UDM-Predictor, deleterious in CADD and PROVEAN; but tolerated in SIFT and Fathmm, neutral in SNP&GO, PhD-SNP and SNAP, unknown in PON-P2, and low impact in Mutation Assessor (Table 2). Variant c.2987G>A (p.C996Y) is not reported in the gnomAD, 1000G, ExAC, Vietnamese ethnic, and in-house database (*n* = 80). This variant was predicted to be high impact, damaging and pathogenic in 15 tools with high scores (Table 2). Both variants can be classified as “*likely pathogenic*” according to the ACMG criteria (Table 2 and Appendix A).

### 2.6. Amino Acid Sequence Alignment and Protein Analyses

For each of the candidate variants, multiple protein sequence alignment was performed using ClustalW (http://embnet.vital-it.ch/software/ClustalW.html). The LAMA2 protein domains have been obtained by Pfam v. 32.0 (https://pfam.xfam.org/) with UniProt code P24043. The disulfide bonds of wild type and mutant LAMA2 were scanned by ScanProsite (https://prosite.expasy.org/scanprosite/). Several parameters like bulkiness, polarity (Grantham), and hydrophobicity (Kyte & Doolittle) of protein were predicted using ProtScale tool in expasy platform (https://web.expasy.org/protscale/).

Amino acid alignment showed that H260 and C996 are highly conserved across species; therefore, substitutions of H260Y and C996Y may have an impact on normal enzyme activity (Figure 2b). H260Y is located in the laminin N-terminal domain corresponding to residues 33–285 of LAMA2 (Figure 2c). C996Y is located in the laminin EGF-like domain 10 corresponding to residues 967–1013 of LAMA2 (Figure 2c). The ScanProsite predicted the eight cysteine-rich repeats, including C967, C969, C979, C985, C987, C996, C999, and C1011, in the laminin EGF-like domain 10, in which disulfide bonds form between C967 and C979, C969 and C985, C987 and C996, and C999 and C1011 (Appendix A). The C996Y mutation disrupted disulfide bonds between C987 and C996 (Appendix A).

Bulkiness, hydrophobicity, and polarity scores were averaged over a nine-residue window size. Both H260Y and C996Y mutations affected significantly the bulkiness of several amino acids nearby (Appendix A). Bulkiness scores of H260Y (15.177) and C996Y (13.282) mutations increased compared to normal residues H260 (15.659) and C996 (12.774), respectively. Bulkiness score of D262 in mutant LAMA2 elevated significantly (15.933) in comparison to the one in normal LAMA2 (14.549) (Appendix A). Histidine and tyrosine are neutral residues of hydropathy, while cysteine is a hydrophobic residue. Therefore, hydrophobicity of H260Y mutation and wild type H260 were not different significantly (Appendix A), while hydrophobicity of C996Y mutation (−1.544) was decreased in comparison to the wild type C996 (−1.122) (Appendix A). In terms of polarity, histidine is a positively charged residue, while tyrosine and cysteine are uncharged residues. Therefore, polarity of H260Y mutation (8.189) had a lower score than wild type H260 (8.656) (Appendix A), while polarity of C996Y mutation and wild type C996 was not changed significantly (Appendix A).

## 3. Discussion

The two patients presented symptoms of muscular dystrophy such as atrophy, musculoskeletal myopathic face, distal contractures fingers, wide-based stance, motor delay with toe walking, elevated creatine kinase, and white matter lesions in brain [2]. However, these clinical features and laboratory findings were insufficient for making a definitive diagnosis. Genetic testing of other reported diseases in Vietnamese patients, including the *SMN* and *GAA* genes, was performed in the two patients. However, no mutations were detected in these genes. The two patients remained undiagnosed for muscular dystrophies until WES was performed. In this study, after WES and variant filtering, we identified two missense variants, c.778C>T in exon 5 and c.2987G>A in exon 21 of the *LAMA2* gene, in the homozygous state in the two patients. Each mutated allele was inherited from their parents. WES was performed in both siblings and their parents, saving time and labor for the selection and validation of possible pathogenic variants. For example, two siblings carried pathogenic predicted variations in compound heterozygous state in autosomal recessive gene like *TTN* or in heterozygous state in autosomal dominant genes like *BAG3*, *CACNA1G*, *DMPK*, *FAT2*, *KCNA5*, *NEFH*, *ORAI1*, and *SYNE1*, while their parents are asymptomatic carriers of these variants; therefore, we excluded them during variant filtering (Appendix A).

The incidence of *LAMA2*-related muscular dystrophy in Vietnamese population is still underestimated. To date, one Kinh ethnic individual was reported with compound heterozygosity for *LAMA2* variants, including a missense variant c.1964T>C (p.L655P) and a splice site variant c.3556-13T>A [40]. Our study is the second report of *LAMA*2 pathogenic variants in Vietnamese patients; however, our study identified a novel missense mutation, c.2987G>A (pC996Y), in the homozygous state in the two patients with different ethnic from that of the previous study.

Pathogenic variants in the *LAMA2* gene result in an early-onset *LAMA2*-related muscular dystrophy (merosin-deficient congenital muscular dystrophy type 1A (MDC1A; MIM # 607855)) or late-onset *LAMA2*-related muscular dystrophy (limb-girdle muscular dystrophy-23 (LGMD R23; MIM # 618138)) [18]. The two siblings achieved independent walking, indicating late-onset of *LAMA2*-related muscular dystrophy. The two siblings also had other typical symptoms of late-onset *LAMA2*-related muscular dystrophy like proximal muscle weakness (positive Gowers’ sign), delayed motor milestones, elevated CK level, abnormal white matter on brain MRI, normal respiratory function, absence of cardiac involvement, and normal digestive system.

Interestingly, our patients showed toe walking which was observed in a Turkish origin male at 10 year-old with *LAMA2*-related muscular dystrophy [41]. This patient was initially diagnosed with Bethlem myopathy or Emery–Dreifuss muscular dystrophy and screened with the main candidate genes; however, no mutation was identified. A targeted capture-DNA sequencing, including forty-six genes involved in cardiomyopathies revealed this patient carried a nonsense variant, c.4936G>T (p.E1646*), in the *LAMA2* gene. In our study, by using WES, we identified two *LAMA2* missense variants in two patients. However, patient 1 showed a progressive muscle weakness with inability to walk from 9 years of age, which is more severe than the Turkish male in the study of Nelson et al. [41].

The two siblings carried missense variants and achieved independent walking. This genotype–phenotype correlation is consistent with the findings of the previous studies [11,25,42], in which missense variants may cause late-onset *LAMA2*-related muscular dystrophy. To date, 345 disease-associated variants have been reported in the *LAMA2* gene, of which, missense variants account for a minority (46/345, 13.33%) (https://databases.lovd.nl/shared/variants/LAMA2/unique; accessed July 2020). Our findings expand the spectrum of missense variants in *LAMA2*. The two missense variants detected in this study are located most in the regions consisting missense variants, the laminin N-terminal domain, and the EGF-like domain [25].

Variant c.778C>T substitutes tyrosine (Y) for histidine (H) at position 260 in the laminin N-terminal domain (Figure 2c). H260Y changed bulkiness and polarity of several amino acids nearby (Appendix A), leading a possible effect of the local conformation and the folding of the protein. The laminin N-terminal domain of laminin-α2 chain is crucial for laminin-211 polymerization; therefore, p.H260Y may change laminin-211 polymerization [43]. Till now, three missense variants were classified as pathogenic/likely pathogenic in exon 5 of *LAMA2* in the Leiden Open Variation database, including c.713C>A (p.A238D), c.728T>C (p.L243P), and c.812C>T (p.T271I). Of which, only c.713C>A (p.A238D) was identified in the homozygous state in a Southeast Asia boy who presented features of both early-onset and late-onset of merosin deficiency like hypotonia at birth, but the absence of feeding difficulties, limb-girdle pattern of weakness, and partial laminin α-2 deficiency [44]. He also had motor development delay and could walk at the age of 3 years. In addition, his brain MRI indicated white matter hyperintense in the T2W FLAIR and white matter hypointense in the T1W, which were also detected in our patients. The homozygous variant c.778C>T (p.H260Y) is also located in exon 5 and near c.713C>A (p.A238D); therefore, c.778C>T may also affect the function of laminin α-2 in a similar way.

Variant c.2987G>A substitutes the TGT codon of cysteine to the TAT codon for tyrosine at amino acid position 996 (C996Y) in the laminin EGF-like domain 10 (Figure 2c). C996Y affected significantly the bulkiness and hydrophobicity, which may change the local conformation of protein and amino acid side chain packing and protein stability. C996Y also disrupts disulfide bonds between C987 and C996, leading to an abnormal folding of the domain and/or abnormal disulfide bonds with other domains. A substitution of cysteine with arginine was also reported at the residue 996 (C996R) in a Turkish patient who showed partial deficiency of laminin α-2 on the immunocytochemical analysis of muscle biopsy sample [45]. The variant c.2987G>A (p.C996Y) may act in the similar way and may lead to a partial laminin α-2 deficiency. IHC staining of muscular biopsy is a gold standard to identify the expression of laminin α-2; however, in this study, we could not perform muscular biopsy in two patients. It is a limitation of our study. Comparing to the patient in the study of Nissinen et al. [45], who only achieved the ability to stand with support as his maximal motor capacity, our patients obtained an independent walking at five years of age; however, due to a progressive muscle weakness, the proband could not walk at nine years of age.

Pathogenic missense variants in *LAMA2* rarely detected in the homozygous state in the patients. Among 52 pathogenic variants, only one homozygous missense variant, c.2462C>T (p.T821M) was identified in the homozygous state in the two male siblings of an Arab consanguineous parents [42]. The two patients presented with contractures, normal intellect, and abnormal white matter hyperintensities on T2-MRI and obtained sitting as maximum motor milestone. However, the younger brother showed scoliosis and febrile seizure at 3 years of age while the elder brother did not [42]. Kubota et al. [46] reported a pathogenic missense variant, c.818G>A (p.Arg273Lys), in the homozygous state in a male patient of consanguineous Japanese parents. This patient could walk at y year of age. He showed an elevated CK level and myopathic change. He was diagnosed with progressive muscular dystrophy. He showed epilepsy at 18 years of age and has been treated with antiepileptic drugs. At age 26 years, he showed diffuse muscular atrophy, mild weakness of masseter and sternocleidomastoid muscles, joint contracture, and deformity. His CK level became normal. He could walk independently. MRI revealed leukoencephalopathy and lissencephaly in the temporal and occipital lobes. IHC indicated partial merosin deficiency. Another pathogenic missense variant c.7881T>G (p.H2627Q) was reported in the homozygous state in a Kenyan family of Asian descent with 5 patients presenting different phenotypes [11]. Three patients never achieved walking ability, while one patient obtained ambulation at 4 years of age. Another patient could walk until 10 years of age, however, later she depended on a wheelchair. All patients had problem in feeding and contractures. Three of them showed scoliosis. Merosin was absent in all patients. Hashemi-Gorji et al. [47] identified a likely pathogenic missense variant, c.8665G>A (p.G2889R), in two unrelated Iranian boys of consanguineous parents. Both patients presented with a hypotonia since birth and elevated CK and aldolase in the first year of life. IHC test revealed merosin positive CMD. Brain MRI showed normal conditions before age 1 year. One patient had kyphosis and contractures of elbow and wrist at age 7 years. Another patient also had kyphosis and normal cognitive function at age 6 years [47]. Recently, two homozygous variants, c.2882G>A (p.A961T) and c.4406G>A (p.C1469Y), were reported in a Chinese patient who presented with seizures, slight weakness of the proximal leg muscles, mild cognitive impairment, and severe leukoencephalopathy [48]. Expression of α-dystroglycan and β-dystroglycan was reduced; however, the expression of laminin-α2 was normal [48]. Such results indicated that pathogenic missense variants were associated with variable phenotype from mild to severe *LAMA2*-related MD with normal or partial loss or absence of laminin-α2. In addition, the patients who carry the same pathogenic missense variants still present different phenotypes of laminin-α2. In our study, the younger sister showed ambulation earlier than the elder brother.

In conclusion, we identified two likely pathogenic missense variants in *LAMA2* in two siblings by using WES. The clinical phenotype of both siblings fit with a form of muscular dystrophy related to missense variants affecting *LAMA2* gene. These results lead an accurate diagnosis of both siblings with *LAMA2*-related muscular dystrophy. Our findings demonstrate WES is a useful diagnostic tool for unidentified muscular dystrophies. Further studies are needed to clarify the effect of variants H260Y and C996Y on functions of LAMA2 protein.

## Figures and Tables

**Figure 1 diagnostics-10-00741-f001:**
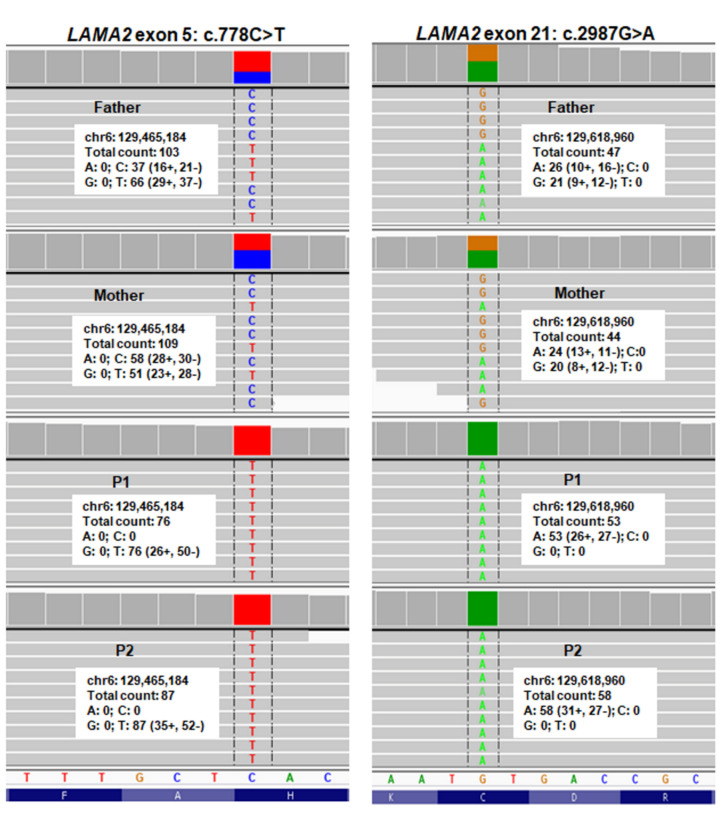
Whole genome sequencing paired-end reads of the family are loaded in the Integrative Genomics Viewer genome browsers.

**Figure 2 diagnostics-10-00741-f002:**
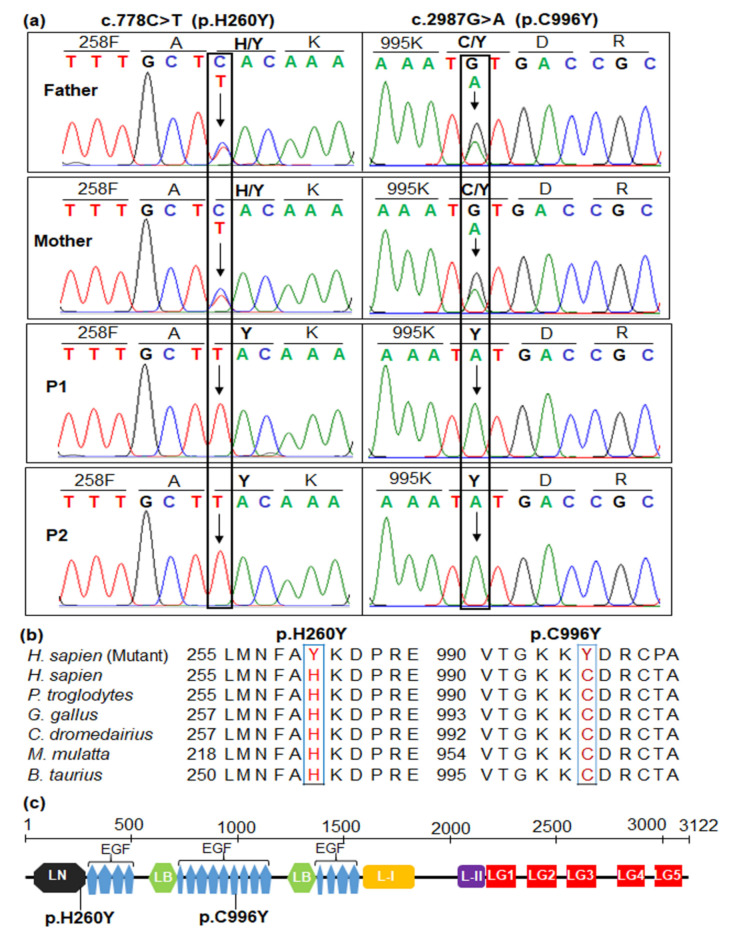
Sanger sequencing validation for two *LAMA2* variants in the family (**a**). Patients P1 (proband) and P2 (younger sister) carried two missense variants, c.778C>T in exon 5 and c.2987G>A in exon 21, of the *LAMA2* gene, in the homozygous state. These variants were inherited from their parents who carried those in the heterozygous state. (**b**) Evolutionary conservation of p.H260Y and p.C996Y. Amino acid alignments indicated that H260 and C996 are highly conserved across species. (**c**) Location of the two variants in LAMA2 protein. LAMA2 includes laminin N-terminal (LN, black hexagon), 17 laminin epidermal growth factor (EGF)-like domains (EGF, blue pentagons), 2 laminin B (LB, light green hexagons), laminin I (I, brown rectangle), laminin II (II, purple rectangle), and 5 laminin globular (LG, red rectangles) (https://pfam.xfam.org/protein/P24043). LAMA2 p.H260Y is mapped in the laminin N-terminal and p.C996Y located in the laminin EGF-10.

**Table 1 diagnostics-10-00741-t001:** Clinical and laboratory data of two siblings.

Patients	P1	P2
Gender	Male	Female
Age of first admission	6-year-old	4-year-old
CK (UI/L) (Normal range: 24–229)	942	523
AST (UI/L) (Normal range: 15–55)	33.9	40.3
ALT (UI/L) (Normal range: 5–40)	10.1	32.7
Head and Neck	Myopathic face Open mouth Prominent jaw Bifid uvula	Myopathic face Open mouth Prominent jaw Broad uvula
Skeletal system	Hyperlordosis A wide-based stance Distal contractures of the fingers	Hyperlordosis A wide-based gait absent reflexes Distal contractures of the fingers
Musculature	Muscular atrophy Macroglossia Motor delay Positive Gowers sign No pseudohypertrophy of tongue and calf muscles Independent walking at 5 years of age Toe walking from 6 years of age Inability to walk from 9 years of age	Muscular atrophy Macroglossia Motor delay Positive Gowers sign No pseudohypertrophy of tongue and calf muscles Independent walking at 4 years of age Toe walking from 5 years of age
Growth development	Delay Normal verbal cognition	Delay Normal verbal cognition
Nervous system	Diffuse white matter changes: hyperintense on T2W and FLAIR sequences, hypointense on T1W sequence Not restricted on diffusion with symmetric appearance Normal corpus callosum, ventricular, pontocerebellar angle, cerebral nerves, medulla, pons and high cervical spinal cord. No fluid collection in meningeal space. No history of seizure	Diffuse white matter changes: hyperintense on T2W and FLAIR sequences, hypointense on T1W sequence Not restricted on diffusion with symmetric appearance Normal corpus callosum, ventricular, pontocerebellar angle, cerebral nerves, medulla, pons and high cervical spinal cord. No fluid collection in meningeal space. No history of seizure
Heart ultrasound	Normal	Normal
Respiratory system	Normal	Normal
Digestive system	Normal	Normal
Karyotype	46, XY	46, XX
Screening pathogenic variants in the *SMN* and *GAA* genes	Negative	Negative

ALT, alanine aminotransferase; AST, aspartate aminotransferase; CK, creatine kinase; FLAIR, fluid-attenuated inversion recovery; MRI, magnetic resonance imaging.

**Table 2 diagnostics-10-00741-t002:** In silico predicted effects of the two candidate missense variants c.778C>T and c.2987G>A in the *LAMA2* gene.

Variant	c.778C>T (p.H260Y)	c.2987G>A (p.C996Y)
	Prediction (Score)	Prediction (Score)
Align-GVGD	Pathogenic (C65)	Pathogenic (C65)
CADD (Phred-scaled score)	Likely deleterious (24.1)	Likely deleterious (29.7)
Fathmm	Tolerated (1.41)	Damaging (−3.40)
Mutation Taster	Disease causing (1)	Disease causing (1)
Mutation Assessor	Low impact (0.9)	High impact (4.14)
MutPred	Pathogenicity (0.822)	Pathogenicity (0.985)
PANTHER	Possibly damaging (361)	Probably damaging (1036)
PhD-SNP	Neutral (RI 6)	Disease (RI 7)
PON-P2	Unknown (0.272)	Pathogenic (0.973)
PolyPhen-2	Damaging (0.988)	Damaging (0.998)
PROVEAN	Deleterious (−3.247)	Deleterious (−10.385)
SIFT	Tolerated (0.148)	Damaging (0)
SNP&GO	Neutral (RI 5)	Disease (RI 9)
SNAP	Neutral (−32)	Effect (53)
UMD-Predictor	Pathogenic (78)	Pathogenic (100)
ACMG classification	Likely pathogenic (PM2, PP1–PP4)	Likely pathogenic (PM2, PM5, PP1–PP4)

PM: pathogenic moderate; PP, pathogenic supporting.

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
