# Peer review of "Whole Exome Sequencing as a Diagnostic Tool for Unidentified Muscular Dystrophy in a Vietnamese Family"

_diagnostics, 2020, doi:10.3390/diagnostics10100741_

Round 1
Reviewer 1 Report
The AA report the case of two Vietnamese siblings presenting clinical symptoms and signs of an unidentified muscular dystrophy, in which a correct diagnosis of LAMA-2 related congenital muscular dystrophy, was reached thanks to the WES.
Both patients presented myopathic face, open mouth, macroglossia, prominent jaw, growth, and motor delay, hyperlordosis, white matter lesions, increased CK values.
Congenital muscular dystrophies are a heterogeneous group of diseases, with subtypes caused by at least 20 different genes. The phenotypes may vary from very severe and fatal forms in the first years of life, to forms with a clinical picture similar to that of a Limb-Girdle Muscular Dystrophy.
The older patient was first tested for DMD, SMN, and GAA genes, probably because muscular dystrophies caused by mutations in these genes are better studied in Vietnam.
Subsequently DNA samples from patients and their parents were analyzed by WES. The analysis showed that both patients share 2 missense variations in homozygous state, in LAMA2 gene. The variations had been inherited from the parents who had them in heterozygosity. One of the variants is novel and never described.
The whole process of analysis and confirmation of the pathogenicity of the identified variants is well described. The references are updated.
I have only one observation to make: the manuscript contains numerous spelling errors, which require review by a native speaker.
Author Response
Thank you for your thorough and positive comments. The manuscript has been reviewed by Professor Garry Warne.
Reviewer 2 Report
The paper describes the case of two siblings suffering from a myopathic process from the very early stage of life; the sons of apparently NON consanguineous parents, were shown carrying two different missense mutations on the LAMA-2 gene. They DO NOT BRING MUTATIONS AT LAMA-2 GENE IN A HOMOZYGOUS STATE; as a matter of fact, both patients are heterozigous compounds for thwo missense mutations on the LAMA-2 gene. Please amend throughout the text accordingly.
Backgroung and introduction should be expanded in order to provide a broader overview of myopathies and muscular distrophies, in particular congenital forms.
The clinical features of both siblings should be better presented as well as the white matter brain MRI alterations.
Sadly, muscle biopsy was not performed; it would have been interesting to look at the pictures; since the clinical phenotype is in keeping with a partial merosin reduction form, muscle biopsy whould likely have shown a reduction of the expression of laminin alpha-2; in such a case, muscle biopsy is extremely useful to point the right diagnosis.
Also, the suspicion for the patients to be affected from Duchenne is clearly wrong; DMD is an X-recessive form of muscular dystrophy and it is absolutely unusual that a female (patient 2) may show such a phenotyle! Please change the text accordingly.
Conclusion should be expanded; clinical phenotypes of both sibs fit with a form of muscular distrophy related to two different missense mutations affecting LAMA-2 gene; it could be nice to discuss briefly the rare cases already reported in literature.
Author Response
We thank you for reviewer’s insightful suggestions. We have revised the manuscript according and detailed corrections are described below.
Point 1: The paper describes the case of two siblings suffering from a myopathic process from the very early stage of life; the sons of apparently NON consanguineous parents, were shown carrying two different missense mutations on the LAMA2 gene. They DO NOT BRING MUTATIONS AT LAMA2 GENE IN A HOMOZYGOUS STATE; as a matter of fact, both patients are heterozygous compounds for two missense mutations on the LAMA2 gene. Please amend throughout the text accordingly.
Response 1: Dear respected Reviewer, please kindly check again the Sanger sequencing of the family (Figure 2, page: 9) to get a clearer view of this point. The parents carry the heterozygous compounds for two missense mutations but asymptomatic. The two siblings carry double missense mutations at the homozygous state and present symptomatic.
Point 2: Background and introduction should be expanded in order to provide a broader overview of myopathies and muscular dystrophies, in particular congenital forms.
Response 2: The introduction has been revised based on your suggestions. Please see changes in Pages: 2-3.
Point 3: The clinical features of both siblings should be better presented as well as the white matter brain MRI alterations.
Response 3: Thank you for the comment. The clinical presentation has been described in more details. Please see changes in Lines: 155-187.
Point 4: Sadly, muscle biopsy was not performed; it would have been interesting to look at the pictures; since the clinical phenotype is in keeping with a partial merosin reduction form, muscle biopsy would likely have shown a reduction of the expression of laminin alpha-2; in such a case, muscle biopsy is extremely useful to point the right diagnosis.
Response 4: We totally agree with your point. However, unfortunately the parents refused to carry out muscle biopsy.
Point 5: Also, the suspicion for the patients to be affected from Duchenne is clearly wrong; DMD is an X-recessive form of muscular dystrophy and it is absolutely unusual that a female (patient 2) may show such a phenotype! Please change the text accordingly.
Response 5: Thank you for noticing this, we have removed this part of the manuscript.
Point 6: Conclusion should be expanded; clinical phenotypes of both sibs fit with a form of muscular dystrophy related to two different missense mutations affecting LAMA2 gene
Response 6: We have amended the conclusion as suggested. Please see change in Lines: 446-447.
Point 7: It could be nice to discuss briefly the rare cases already reported in literature.
Response 7: We have added a brief discussion of pathogenic missense variants identified in the LAMA2 gene. Please see changes in Lines: 413-444.